# Measuring constitutional preferences: A new method for analyzing public consultation data

**Andrés Cruz[1]☯, Zachary Elkins[1]☯, Roy Gardner[2]☯, Matthew Martin[1]☯ \*, Ashley Moran[1]☯**

1 Department of Government, University of Texas at Austin, Austin, Texas, United States of America,
2 Comparative Constitutions Project, University of Texas at Austin, Austin, Texas, United States of America

☯ These authors contributed equally to this work.
\* mjmartin@utexas.edu

## Abstract

Public consultation has become an indispensable part of constitutional design, yet the voluminous, narrative data produced are often impractical to analyze. There are also few, if any, standards for such analysis. Using a comprehensive reference ontology from the Comparative Constitutions Project (CCP), we develop a new methodology to identify constitutional topics of most concern to citizens and compare these to topics in constitutions globally. We analyze data from Chile's 2016 public consultations—an ambitious process that produced nearly 265,000 narrative responses and launched the constitutional reform process that remains underway today. We leverage advances in natural language processing, in particular sentence-level semantic similarity technology, to classify consultation responses with respect to constitutional topics. Our methodology has potential for advocates, drafters, and researchers seeking to analyze public consultation data that too often go unexamined.

## Introduction

Constitutional reform processes often begin by consulting citizens about their preferences, concerns, and priorities. These consultations typically solicit open-ended public comments, an unstructured approach that presents significant analytic challenges [1–3]. Such public input generates volumes of text [4] that are often unwieldy to analyze and inaccessible to policy makers and interest groups. For example, preceding the 1987–1988 Brazilian Constituent Assembly, the federal government asked citizens to submit proposals by postcard, resulting in a mountain of 72,719 postcards that looked impressive in photos but was of limited use to delegates [5]. The challenge is widespread; Horowitz notes that public consultations drew over 25,000 suggestions in Uganda in 1995, nearly 40,000 in Kenya in 2009, and over 70,000 in Ghana in 2010 [6]. Yet there are few standards for how to analyze or represent the suggestions produced by constitutional consultations, and research on such consultations remains sparse.

Our general research objective is one we call *conceptual ecology*, meaning the identification and inventory of ideas in varied contexts. A study of conceptual ecology might assess the prevalence of ideas in a particular time or place (the *Zeitgeist*), or the ideas in a particular collection of documents. In the case we present here, we explore both, analyzing the ideas surfaced by a national campaign for public comment on constitutional reform and comparing these to the

**Data Availability Statement:** All data used in this study are available in a repository on Zenodo (https://doi.org/10.5281/zenodo.10206513).

**Funding:** This work was supported by the National Science Foundation award number 2315189,

awarded to ZE and AM. The funders had no role in study design, data collection and analysis, decision to publish, or preparation of the manuscript.

**Competing interests:** The authors have declared that no competing interests exist.

ideas in all current national constitutions globally. A common analytic approach to discovering ideas in a large corpus is *topic modeling*, an inductive approach that can help cluster topics raised in such consultations and reduce the dimensionality of ideas in the corpus. Skillful interpretation of the clusters *can* lead to topic classification and appropriate descriptive inferences, especially in exploratory phases of research [7]. However, a more confirmatory approach with an *a priori* set of topics is potentially more amenable to testing theoretical expectations.

Our approach is more deductive in this sense, and relies upon a well-developed ontology (or, vocabulary) of constitutional topics combined with recent advances in text-similarity measurement. Starting with a comprehensive ontology enables an assessment of both what citizens suggest and what they do *not* suggest, and it also enables systematic comparison across populations. To develop our approach, we analyze public consultation data from Chile, gathered at the outset of a continuing effort to replace the constitution drafted under a dictatorship over four decades ago. Chile's 2016 consultation process was similar to that of other contemporary constitutional projects, with the government organizing a structured process for public input into constitutional reforms. Given the regional influence of Chile and the historic nature of the reform, the country's process may be something of a model case. The public preferences expressed in Chile's well-structured consultation process present a wealth of systematic data for analysis. As is the case for many such processes, these data have received strikingly little attention.

Critical to any study that seeks to analyze public preferences is the point of comparison. In this analysis, we compare the ideas of Chilean citizens to concepts in constitutions around the world. This enables us to measure not only what Chileans wish to see in their constitution, but also how these ideas align with those already in circulation in the world's constitutions.

## Materials and methods

### Research questions

We introduce a set of methods for the analysis of a range of substantive and structural aspects of public consultations. As we detail in the data section below, the Chilean consultations asked participants for their views on the most important values, rights, duties, and institutions to be included in a new constitution, collecting a range of ideas about the potential substance of the constitution. This input was collected in a three-stage consultation process designed by the Chilean government. Each stage used a similar format but different convening mechanisms. At the local level, consultations were run by community groups; at the provincial level they were run by provincial governors; and at the regional level they were run by regional administrators. Importantly, these consultations were conducted nationwide, collecting input from people living in widely varying residential contexts across urban and rural areas.

We structure our research questions to analyze the following three sets of substantive and structural issues that arise from the Chilean public consultations. Our particular research questions, which would be relevant to constitutional drafters, address:

1. **Idea Representation**. To measure the constitutional preferences of Chilean participants, we ask: How do the ideas suggested by Chilean participants compare with the universe of ideas in the world's constitutions? Are the topics most frequently included in constitutions also those most frequently mentioned in Chilean consultation responses? If not, where do constitutions and Chilean consultation responses align and differ in their most prevalent topics?

2. **Consultation Structure**. Since the data collection process was somewhat different at each administrative level where consultations were conducted (i.e., at the local, provincial, and

regional levels), analysts might wonder about artifacts of the process. To assess the effects of different data collection methods on participant preferences, we ask: What is the relationship between topics raised by participants and the administrative level at which consultations were conducted?

3. **Residential Context**. Since people's experiences and challenges can differ in the varied residential contexts where consultation participants live (i.e., urban, rural, and mixed municipalities), analysts might wonder about their effects on participant responses. To assess the effects of different residential contexts on participant preferences, we ask: What is the relationship between topics raised by participants and the type of municipality where consultation participants live?

These questions would be intractable without an ontology against which to compare constitutional and consultation content, and a method for analyzing vast volumes of text. An ontology (related to taxonomy, schema, and controlled vocabulary) helps to structure knowledge in a particular domain by specifying a comprehensive set of concepts related to that domain. By identifying a universe of ideas, a comprehensive ontology helps to provide a larger perspective on smaller samples of ideas from different populations. An analogy might be a naturalist who assesses the diversity of plants in an ecosystem by comparing a sample of specimens from that ecosystem to a complete catalog of the world's plants.

Here, we introduce a constitutional ontology designed to index the conceptual content of the world's constitutions, and we use semantic similarity technology to align the ontology with ideas in both consultation responses and constitutional texts. We are thus able to compare the preferences of Chilean citizens with the concepts embodied in national constitutions from across the globe. However, we do not expect the consultations to be perfectly representative of the ideas in circulation in constitutions. The core question is, which ideas surface and how different are they from those in national constitutions globally? Below we present a method for identifying, analyzing, and comparing topics in the unstructured data produced by constitutional consultations. The method is also applicable to exploring large volumes of text generated by other processes.

## Data

**Reference ontology.** Our reference ontology is an elaboration of the ontology developed by the Comparative Constitutions Project (CCP). CCP's authors, Elkins and Ginsburg, track the content of nearly all of the world's historical and current constitutions—and their amendments—since 1789 [8]. CCP also indexes the texts of in-force constitutions across 330 topics and, in a collaboration with Google, displays these texts on Constitute, a site intended to assist constitution drafting. Constitutional drafters, citizens, and scholars can thus access sections of constitutional texts related to these topics on the website. To track these topics in constitutions, Elkins and Ginsburg developed the CCP ontology as a set of concepts and categories to capture systematically the content of constitutions. They intend this topic set to be highly inclusive, since their research questions rely upon a fairly complete inventory of constitutional ideas. Nevertheless, their topic set is necessarily incomplete—constitutional innovations appear each year and some of the more idiosyncratic constitutional ideas are not tracked. Also, other researchers would conceptualize topics in different ways, privileging different characteristics. Nevertheless, the ontology serves as a useful baseline of the world's constitutional ideas, however incomplete. We use this set of CCP topics, make modifications to increase the meaningful semantic distance between topics for improved use with semantic-similarity tools, and add a few topics from the broader CCP dataset.

Each topic comprises a label field and a longer description field. To prepare topics for encoding, we generate grammatical (or near-grammatical) multi-sentence text from these topic fields. For example, the text segment for the topic on freedom of movement is structured as follows: "*Freedom of Movement. Grants citizens the right to travel and reside anywhere in the state, and to travel to and from other countries.*" Similarly, the segment for the topic on the separation of church and state reads as follows: "*Separation of church and state. Guarantees a secular state, guarantees a separation between the state and religious organizations, or forbids the state from extending recognition or assistance to religious organizations.*" In total, our ontology includes 334 English-language topics. We use this set of 334 constitutional topics as the set of sentence-level phrases needed to retrieve semantically similar sentence-level text from our textual data sources.

**Textual data sources.** We measure the semantic similarities between topics in the CCP ontology and text segments from two corpora: a corpus of consultation responses collected in Chile in 2016, and a corpus of current national constitutions. In other words, we classify consultation responses and constitution sections by measuring their similarity to CCP topics. We then compare the distribution of topics across consultation responses with that across constitution sections. The Chilean consultation responses were collected by the Chilean government in 2016 as part of a constitutional reform process introduced by former President Michelle Bachelet to replace the 1980 constitution. That constitution was drafted by a small team of elite lawyers during the military dictatorship of General Augusto Pinochet to entrench the regime's political interests. Replacing the Pinochet constitution presented both an answer to the legacies of social and political exclusion and, for some Chileans, a long-awaited disavowal of the military era [9]. The process featured a series of direct consultations to gather the constitutional preferences of Chilean citizens. The Chilean government then anonymized and published the consultation data in 2017. Ultimately, however, Bachelet's constitutional project stalled following the election of conservative President Sebastián Piñera in 2017. Although the promise of constitutional reform was not revived until the country's estallido social ("social outbreak") in 2019, the public consultations held during the Bachelet process had reserved a wealth of systematic data for analysis.

All told, 218,689 Chileans participated across four stages of the consultation process in 2016 —about 1.2% of Chile's population at the time [10]. More than 90,000 citizens submitted an individual questionnaire, and another 127,000 took part in an in-person, deliberative event held at the local, provincial, or regional level [11]. We are interested in the textual data generated by the three stages of group deliberations that occurred at the local, provincial, and regional events. The first deliberative stage consisted of a series of relatively uncoordinated focus groups—*encuentros locales autoconvocados* ('local self-convened encounters,' or ELAs), which occurred from April to June 2016. These meetings were initiated and led by citizens and non-governmental organizations. The second and third stages that followed were town-hall meetings—*cabildos*—organized by provincial and regional government officials from July to August 2016 [12].

Each stage of the consultation had a similar, though not identical, structure. In all three deliberative stages, participants responded to queries regarding four aspects of constitutional reform: (a) values and principles; (b) rights; (c) duties and responsibilities; and (d) state institutions. For each aspect, participants were required to answer one semi-structured question that probed their preferences for a new constitutional text. The ELAs also included a fifth, optional, open-ended question that invited participants to offer a comment as part of the "historical memory" of the constitutional reform process [11].

For the semi-structured questions, participants were asked to identify the most important concepts to be included in the constitution in each of the four areas (values, rights, duties, and

institutions). They could choose from a set of concepts predefined by the organizers or suggest their own original concepts. For the values question, for example, they could choose among 37 predefined concepts such as decentralization, democracy, freedom, and gender equality, or they could suggest their own concepts [12]. For the rights, duties, and institutions questions, there were 44, 12, and 21 predefined concepts, respectively, all of which are available for reference alongside our other data in a repository to support replication of our results. Self-defined concepts sometimes provided more nuance on topics predefined by the government. For example, the predefined right to 'respect for the environment' and one group's self-defined right to 'an environment free of pollution' both relate to the environment, but the latter provides more detailed information about constituent views. At other times, self-defined concepts added wholly new topics to the discussion, as with groups' suggestions for rights to 'conscientious objection' and 'binding public participation in government decisions' [13].

Participants could choose from 114 predefined values, rights, duties, and institutions. Participants suggested 36,292 *additional* self-defined values, rights, duties, and institutions for drafters to consider. These self-defined ideas composed nearly 14% of all suggestions for the most important ideas to include in the new constitution [12]. This underscores both the importance of this opportunity for constituents to express their unique views and the unwieldy amount of unstructured data produced by the consultation process.

Importantly, the consultations were organized such that the information flow was cumulative. In the first stage of group deliberation—the ELAs—participants could list up to seven concepts per question, including predefined and new concepts. The provincial and regional cabildos were then organized to serve as a continuation of the ELAs, as concepts identified as most important by the ELAs were used to inform deliberations in the provincial cabildos. Groups in the provincial cabildos each identified the seven most important concepts identified by the ELAs in each area; the group could then add up to seven *additional* concepts in each of the four areas. This meant each group in each provincial cabildo could identify 7 to 14 topics in each area of values, rights, duties, and institutions as their top priorities for inclusion in the constitution. The same approach was used for the regional cabildos, which used input from the provincial cabildos for their deliberations [12].

The ELAs and provincial and regional cabildos followed similar decision-making procedures. Importantly, preferences were collected at the group level, not as individual submissions, so each response reflects the synthesized views of multiple people. For each concept, the group justified its reasons for choosing that particular concept using participants' own words, known as their *fundamentos* or "bases" for their collective decision [12].

For our purposes, consultation responses include the entire answer provided by the Chilean consultation group—i.e., both the concept and the explanation for why the group chose that concept as being critical for inclusion in the constitution. For example, a group selecting the predefined right to 'respect for the environment' noted above elaborated that this right is important because it related to natural resources and animal rights; the group's full consultation response we use is thus "Respect for the environment. Protection of natural resources during their exploitation, ensuring their added value to the country; Recognition of animals as beings with rights" [13]. Again, then, even for the predefined responses the consultation format enabled participants to share their unique ideas in narrative, expositive form—an important consultative experience and a challenging analytical one. This paper includes consultation responses gathered at the events held at all three administrative levels: ELAs, provincial cabildos, and regional cabildos. In total, there are 264,800 consultation responses.

The constitution corpus comprises 192 constitutions. CCP has segmented these constitutional texts according to their hierarchical structure (sections, subsections, etc.). We flatten the hierarchy (ignoring section information) and analyze only those segments containing the

substantive content of constitution sections (i.e., not titles and headers). Altogether, the 192 national constitutions provide a total of 163,102 constitution sections for analysis.

To access the data for this study, we compiled the national constitutions we maintain for CCP [8] and downloaded the consultation data from the Chilean government's Open Data Portal [14]. All consultation responses used in this study are publicly available online via the Chilean government's portal. Since the consultation data were collected and published by the Chilean government prior to this study, this study's researchers were not involved in obtaining participant consent. All consultation data were fully anonymized by the Chilean government before they were accessed by this study.

## Methodology

Our objective is to index each of the two corpora with topics from the ontology and then compare the topics found in each. Our approach to this classification task is to calculate the semantic similarity between topics in the CCP ontology and constitution sections, and between topics and consultation responses. Sentence-level vector semantic methods measure the degree to which two or more natural language sentences or clauses convey similar meaning. We use version 3 of Google's multilingual Universal Sentence Encoder (USE) to generate numerical representations of topics, constitution sections, and consultation responses. These numerical representations are referred to as encoding vectors or embeddings [15,16]. An encoding vector defines the coordinates of a topic, constitution section, or consultation response in a 512-dimensional semantic space where proximate topics, sections, and responses in the space are considered to have similar meaning.

For USE models, the preferred measurement of the distance between a pair of encoding vectors is angular distance. The inverse of this distance—vector proximity—constitutes our similarity score for each pair of sentences. This measurement is referred to as a semantic similarity score, and the score ranges from 1.0 to 0.0. A similarity score of 1.0 means two sentences are identical in meaning and comprise the same words in the same order. As the meaning of the sentences diverges, the similarity score decreases.

One of the key features of the multilingual USE model is that it enables efficient and accurate computation of sentence-level encoding vectors, making it possible to perform large-scale semantic similarity tasks on a range of datasets in different languages without any pre-processing of text other than segmentation. We measure the semantic similarity between English CCP topics and constitution sections, and between English CCP topics and Spanish consultation responses. To build confidence in our choice of the English version of the CCP ontology, we confirmed that both the English and Spanish versions of the CCP ontology generated almost identical sets of scores across constitution sections.

For constitutions, we build a similarity matrix in which each row corresponds to a CCP topic, and each column represents a constitution section. Cell values contain the semantic similarity score for a topic-section pair. For consultation responses, we build a matrix for each of the three administrative levels. Once again, topics are in rows whereas columns represent consultation responses. Cell values contain the semantic similarity score for a topic-response pair. A row in a matrix can be sorted by similarity score to find the sections or responses that are most similar to a topic.

To reduce the incidence of false positives, we adopt a threshold value of similarity. Our analysis of a sample of topic-sentence pairs from various corpora suggests that a threshold value of 0.7 represents a conservative cutoff, with a minimal number of false positives. As the threshold decreases, the number of false positive results increases. For example, the constitutional text reading "*to approve treaties, conventions and such other international agreements*

*negotiated or signed on behalf of the Republic*" has similarity scores of 0.73 to the truly similar topic "*International Law. Reference to international law, including treaties, agreements, norms, and other commitments,*" 0.62 to the marginally similar topic "*Customary international law. Reference to customary international law, the law of nations, or the generally accepted principles of international law,*" and 0.52 to the dissimilar topic "*Ordinary court term limits. Maximum number of terms that an ordinary court judge serves.*" Our choice of threshold thus errs on the side of excluding somewhat similar—rather than including dissimilar—topic-section and topic-response pairs.

To determine how frequently a topic appears in constitutions and consultation responses, we measure the number of sections and responses that are semantically similar to that topic at or above our similarity threshold of 0.7.

## Results

Using the values in our semantic similarity matrices, we calculated a results matrix with the 334 CCP topics in rows and eight columns—one for constitution sections, one each for consultation responses collected at each of the three administrative levels, one each for consultation responses collected in each of the three municipality types, and one for consultation responses aggregated across all administrative levels and municipality types. Cells in the results matrix contain the number of constitution sections or consultation responses that are semantically similar to a row's topic at or above our threshold of 0.7. Column cell values were then scaled to the range 0 to 1 by dividing each cell value by its column total. Cell values were then converted to percentages to represent how frequently a topic appears in constitutions and consultation responses.

### Top-10 topics

An initial analysis involved using the results matrix to make a simple comparison of the ten most frequently mentioned topics in constitutions compared with those in aggregated consultation responses. If the topics of most interest to Chilean consultation participants align with the most prevalent topics in constitutions, then we might expect some overlap between the top-10 lists. Table 1 suggests very little overlap. Topics relating to the constitutional court and the supreme court rank highest in constitutions, whereas topics relating to equality rank highest in consultation responses. This finding simply reminds us that the core content of a typical constitution is not necessarily related to the ideas advanced by citizens, who are likely motivated to propose new ideas.

**Table 1. Ten most prevalent topics in national constitutions and aggregated Chilean consultation responses.**

| Rank | Constitution Sections | Consultation Responses |
|---|---|---|
| 1 | Constitutional court powers | Equality before the law |
| 2 | Supreme court powers | Equality regardless of gender |
| 3 | Constitutional court | Equality regardless of sexual orientation |
| 4 | Constitutional court removal | Equality regardless of religion |
| 5 | Deputy executive | Democracy |
| 6 | Judicial precedent | Equality regardless of race |
| 7 | Constitutional court selection | Constitutional court powers |
| 8 | Cabinet/ministers | Environment |
| 9 | Electoral court eligibility | Constitution amendment |
| 10 | Supreme court eligibility | Supreme court powers |

## All topics

For a more detailed analysis, we used the results matrix to compare the distribution of consultation responses across all topics with the distribution of constitution sections across all topics. The results are shown in Fig 1. The left-hand side of Fig 1 (in red) shows the ranked percentage of constitution sections that align with each of the 334 CCP topics. Clearly, CCP topics are not equally represented in national constitutions, as some topics feature more than others, and

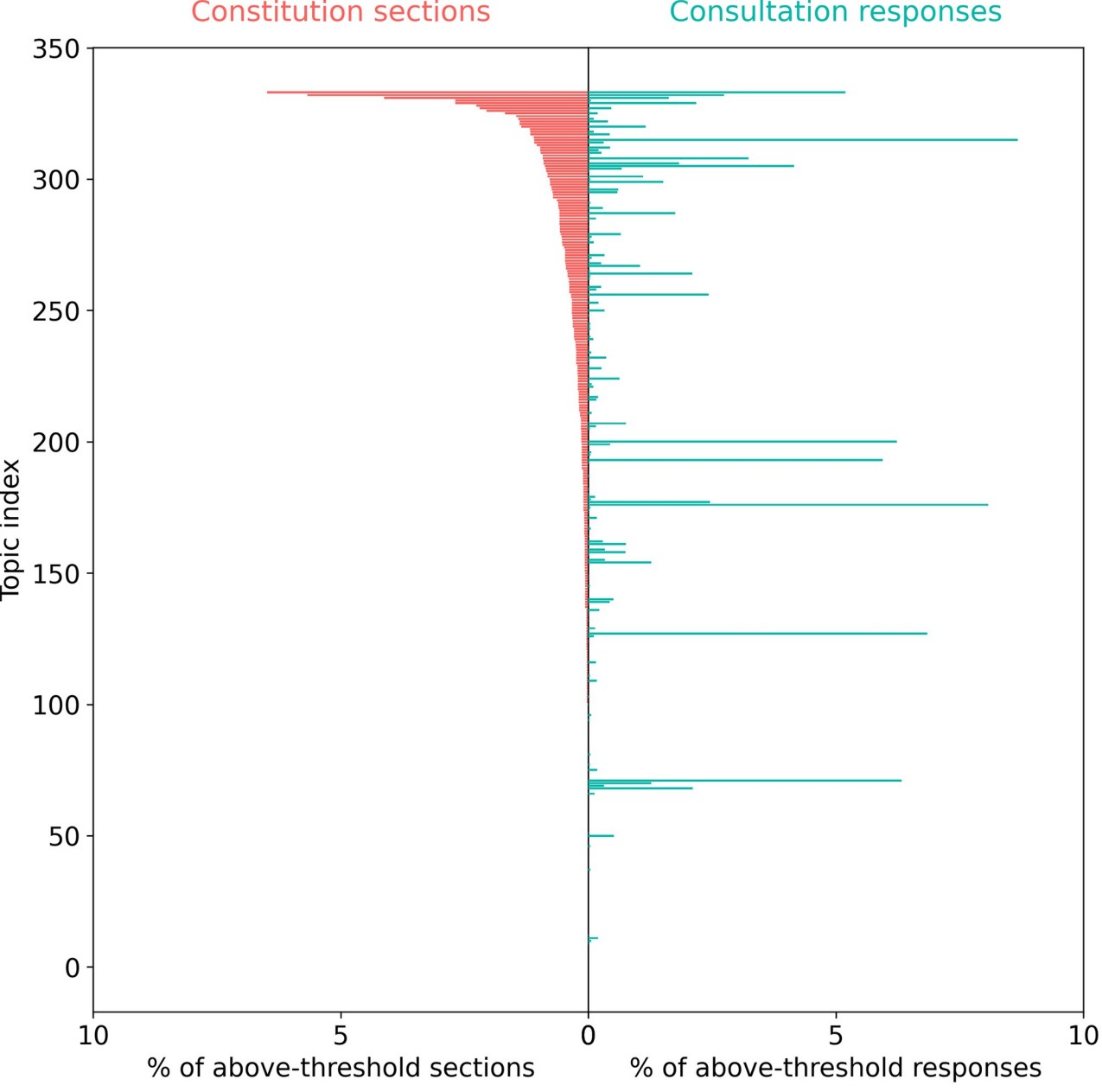

**Fig 1. Comparison of the ranked distribution of constitution sections (red) and aggregated consultation responses (green) for all 334 CCP topics.** The figure shows the number of constitution sections and consultation responses for each topic, expressed as a percentage of the total number of above-threshold sections or responses across all topics. The x-axis shows the percentage of constitution sections and consultation responses. The y-axis shows the CCP topic index.

some are not mentioned at all. The right-hand side of the figure (in green) shows the percentage of Chilean consultation responses that align with each CCP topic. To facilitate comparison, the topic order is the same as that for constitution sections. It can be seen that CCP topics are not equally represented within constitution sections or consultation responses, with some appearing much more frequently than others.

Using Pearson's correlation coefficient, we find a weak positive correlation between the distribution of topics in national constitution sections and the distribution of topics in Chilean consultation responses (r = 0.29, p < 0.001). The results in Fig 1 were obtained using aggregated consultation responses, that is, aggregating responses over the three administrative levels —regional, provincial, ELA; and within provincial and ELA, aggregating over the three different municipality types—urban, rural, mixed.

To determine whether the aggregation of responses is justified, we use the results matrix to compare the distribution of responses from the three administrative levels. Statistical analysis shows that there is a very strong positive correlation between the three administrative levels, as shown in Table 2. We can conclude that the administrative level of consultation does not affect the distribution of responses over topics.

Similarly, statistical analysis shows that there is a very strong positive correlation between the three municipality types as shown in Table 3, i.e., municipality does not affect the distribution of responses over topics.

We can therefore confirm that aggregation of Chilean consultation responses across administrative levels and across municipalities is appropriate.

Fig 2 compares the prevalence of topics in national constitutions and Chilean consultations, aggregated to the twelve categories of constitutional topics, as organized in the CCP ontology. It is clear that consultations in Chile focused much more extensively on rights and duties than do national constitutions. Rights topics compose a significant proportion of content in national constitutions (18%); however, rights topics compose more than double this proportion in consultation responses (45%). Chilean consultation responses also invoke culture and identity topics, principles and symbols topics, and constitutional amendment topics at higher frequencies than constitutions, as Fig 2 shows. Yet consultation responses invoke topics on the judiciary, executive, legislature, elections, and regulatory oversight at a much lower frequency than national constitutions.

## Rights-related topics

One likely explanation of Chilean participants' topical focus is procedural. The consultation agenda was, in part, a function of the structure of the organizers' questions. Recall that consultations solicited participant responses in the four areas of rights, duties, values, and

**Table 2. Pearson's correlation coefficients for comparisons of consultation response topics across administrative levels for all 334 CCP topics.**

| Source | Source | Pearson's |
|---|---|---|
| Regional | Provincial | 0.99 |
| Regional | Local | 0.95 |
| Provincial | Local | 0.96 |

Pearson's correlation coefficients are shown for pairwise comparisons between the distributions of consultation responses over topics for the three administrative levels of consultation. All pairs of distributions are highly correlated with p<0.001.

**Table 3. Pearson's correlation coefficients for comparisons of consultation response topics across municipality types within and across administrative levels for all 334 CCP topics.**

| Source | Source | Pearson's |
|---|---|---|
| Provincial-Urban | Provincial-Rural | 0.96 |
| Provincial-Urban | Provincial-Mixed | 0.93 |
| Provincial-Rural | Provincial-Mixed | 0.95 |
| Local-Urban | Local-Rural | 0.99 |
| Local-Urban | Local-Mixed | 0.99 |
| Local-Rural | Local-Mixed | 0.99 |
| Provincial-Urban | Local-Urban | 0.96 |
| Provincial-Rural | Local-Rural | 0.97 |
| Provincial-Mixed | Local-Mixed | 0.92 |

Pearson's correlation coefficients are shown for pairwise combinations of comparisons between the distributions of consultation responses over topics for municipality types within and across the local and provincial administrative levels. All pairs of distributions are highly correlated with $p < 0.001$.

institutions. It may not be surprising, then, that nearly two thirds of consultation responses fall into CCP categories for rights and duties, culture and identity, and principles, as Fig 2 shows. Given these structural constraints, it is useful to focus the analysis, as much as possible, on comparing the prevalence of topics within each of these categories.

The discussion that follows thus limits the analysis to rights, an area well represented in both constitutional and consultation corpora. For this analysis, we selected all 97 rights topics in the CCP ontology and used these to extract a sub-matrix from our main data matrix. Within this sub-matrix, column cell values were then scaled to the range 0 to 1 by dividing each cell value by its column total. Cell values were then converted to percentages to represent how frequently rights topics are mentioned in sections and responses.

Fig 3 shows the distributions of constitution sections and aggregated consultation responses over the 30 rights topics that are most prevalent in national constitutions. Evidently, national constitutions address a much broader range of rights and duties than consultation participants did. In some ways, this is unsurprising, given the wide variation in constitutional contexts and needs across the globe. However, many rights that are widely entrenched in constitutions globally appear rarely or not at all in above-threshold Chilean consultation responses. These include criminal procedure rights like bans on cruel treatment, unjustified restraint, and false imprisonment; trial rights like the presumption of innocence, ban on punishment outside the law, and right to a speedy trial; and rights to free press, matrimonial equality, education, and employment.

Fig 4 shows the distributions of constitution sections and consultation responses over the 30 rights that are most prevalent in consultation responses. This reveals many rights that are raised at a much higher frequency in Chilean consultation responses than in constitutions globally. These include equality issues (particularly gender equality), protection of the environment, freedom of expression, equal pay for equal work, the right to life, the right to a jury trial, and the rule of law. This could be an indication that these topics are uniquely critical to the current Chilean context.

Statistical analysis supports the idea that consultation participants discussed a different profile of rights, compared with those that appear in current constitutions. Across the 97 rights topics tracked, there is only a weak positive correlation ($r = 0.32$, $p < 0.01$) between the appearance of topics in national constitutions and Chilean consultations.

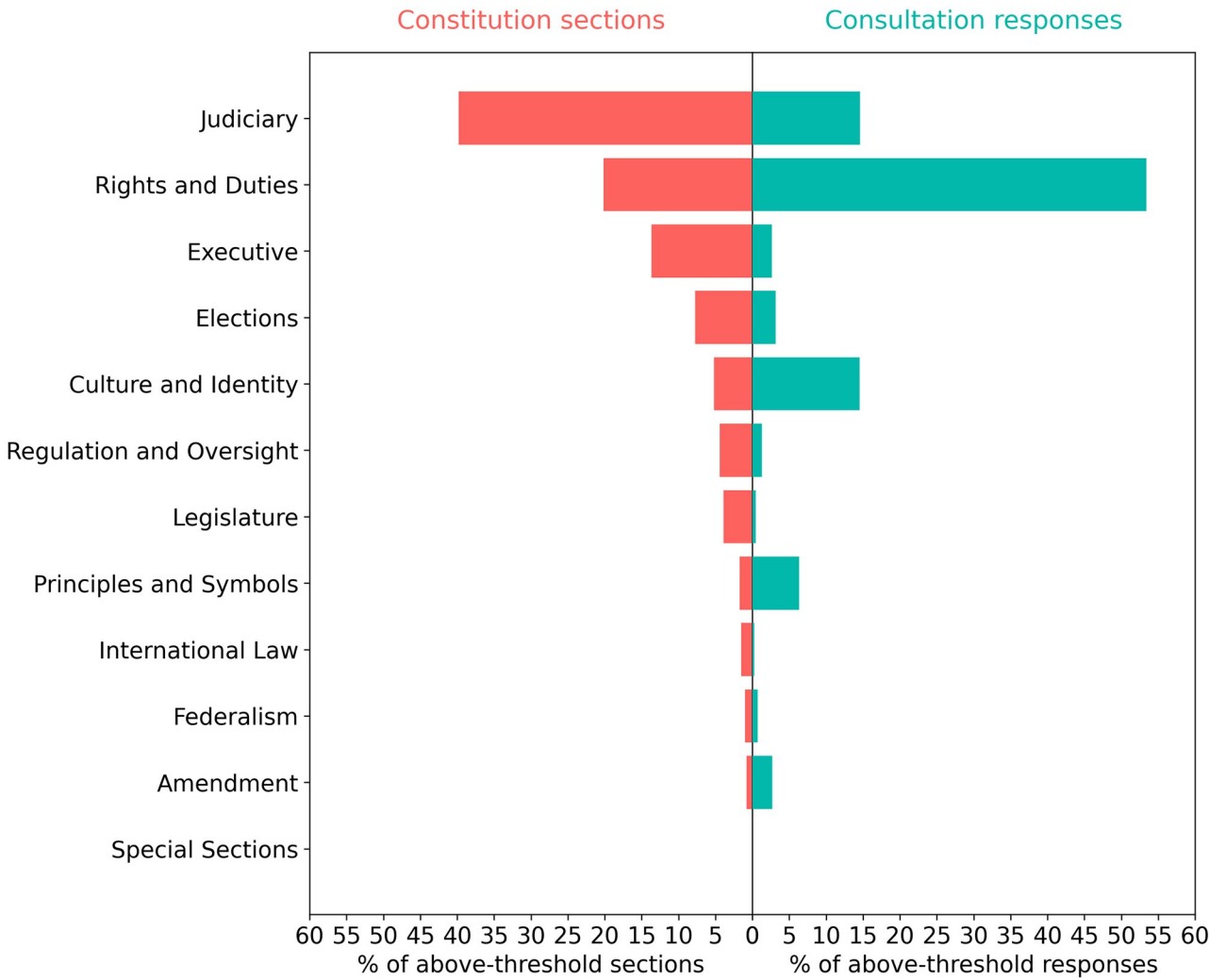

**Fig 2. Comparison of the ranked distribution of constitution sections (red) and aggregated consultation responses (green) across 12 categories of constitutional topics.** The figure shows the number of constitution sections and consultation responses for each topic category, expressed as a percentage of the total number of above-threshold sections or responses across all topics. The x-axis shows the percentage of constitution sections and consultation responses. The y-axis shows topic category labels.

The results in Figs 3 and 4 were obtained using aggregated responses. Statistical analysis shows that within rights topics there is a very strong positive correlation between the three administrative levels and between the three municipalities, as Table 4 shows. We can conclude that aggregation of consultation responses within rights topics is appropriate since neither administrative level nor municipality affects response distribution for this category of topics.

## Institution-related topics

The following discussion restricts the analysis to institution-related topics. To do this, we selected the 184 institution-related topics in the CCP ontology and used these to extract a sub-matrix from our main data matrix. Within this sub-matrix, column cell values were then scaled to the range 0 to 1 by dividing each cell value by its column total. Cell values were then

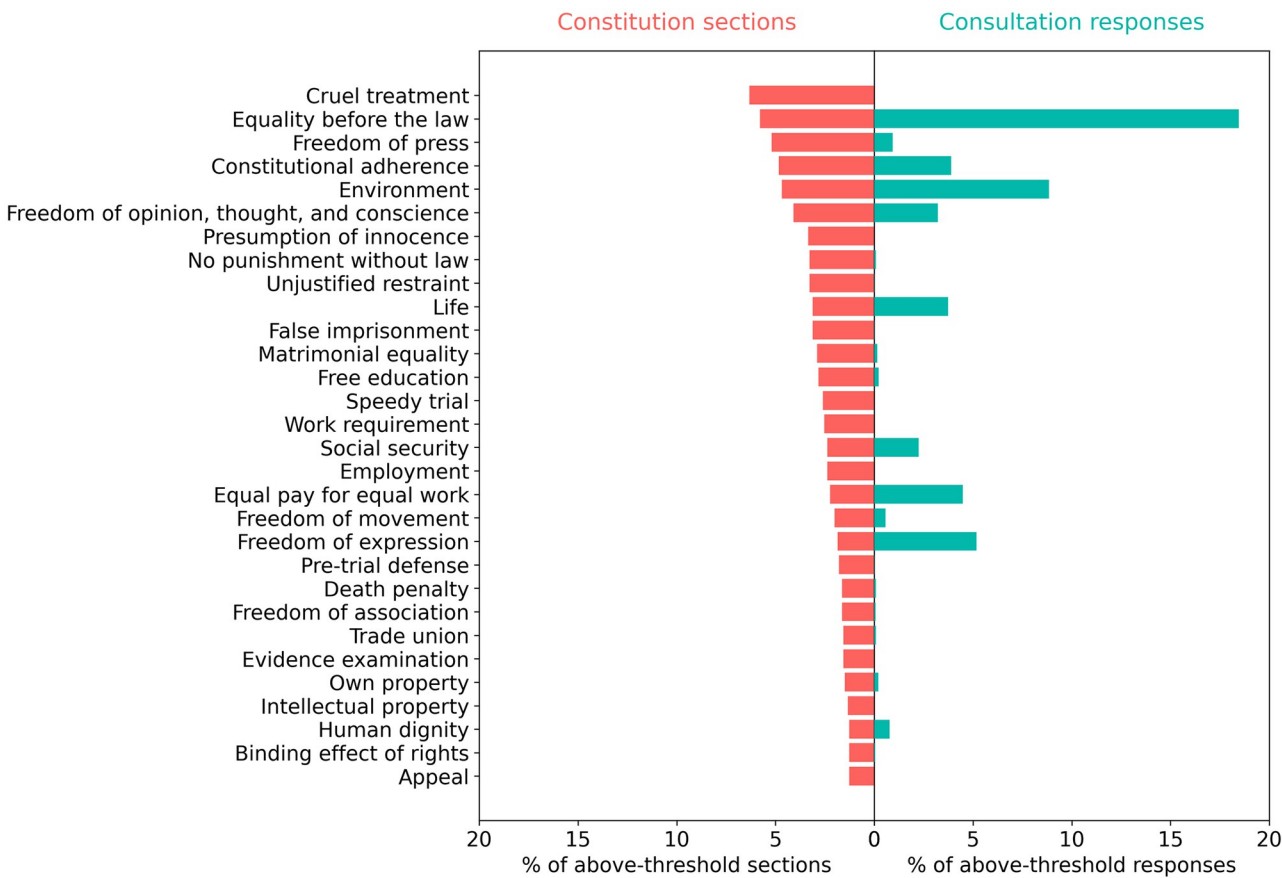

**Fig 3. Comparison of the ranked distribution of constitution sections (red) and aggregated consultation responses (green) across the 30 rights topics most prevalent in national constitutions.** The figure shows the number of constitution sections and consultation responses for each rights topic, expressed as a percentage of the total number of above-threshold sections or responses across all rights topics. The x-axis shows the percentage of constitution sections and consultation responses. The y-axis shows topic labels.

converted to percentages to represent how frequently an institution-related topic is mentioned in sections and responses.

Fig 5 shows the distributions of constitution sections and consultation responses over the 30 institutional topics that are most prevalent in national constitutions. In contrast to rights topics, there is substantial agreement on the most common institutional topics raised in national constitutions and Chilean consultations. The top four institutional topics raised in national constitutions—constitutional court powers, establishment, and removal, and supreme court powers—are among the top five institutional topics raised in Chilean consultation responses. Yet many institutional topics that are widely addressed in the world's constitutions —including requirements related to deputy executives, the council of ministers, judicial precedent, supreme court eligibility and opinions, and the attorney general—appear only rarely if at all in above-threshold Chilean consultation responses. Overall, mentions of executive topics are strikingly low in the consultation responses.

Fig 6 shows the distributions of constitution sections and consultation responses over the 30 institutional topics that are most prevalent in consultation responses. This reveals institutional topics that are raised at a higher frequency in Chilean consultation responses than in constitutions globally, including constitutional court topics, constitutional amendment

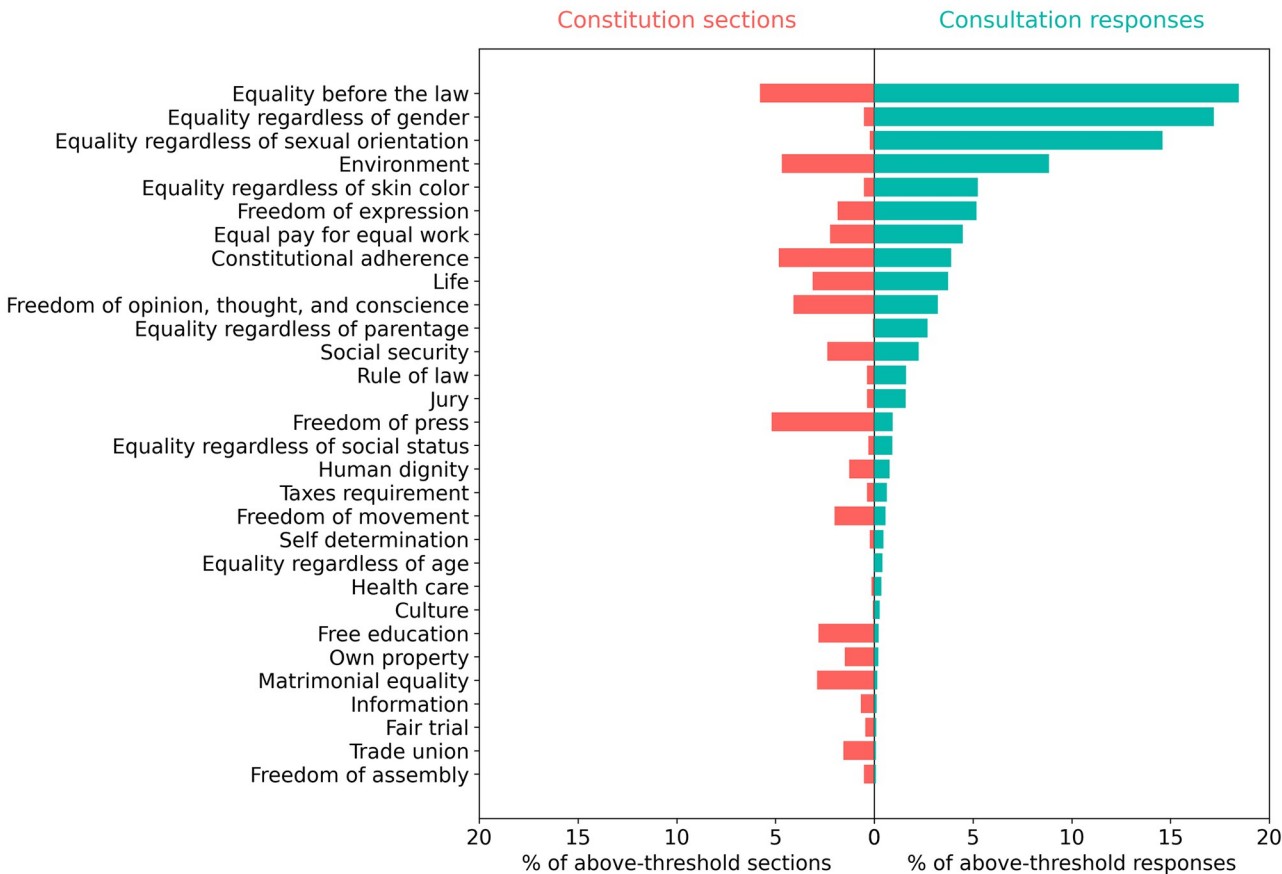

**Fig 4. Comparison of the ranked distribution of consultation responses (green) and aggregated constitution sections (red) across the 30 rights topics that are most prevalent in consultation responses.** The figure shows the number of constitution sections and consultation responses for each rights topic, expressed as a percentage of the total number of above-threshold sections or responses across all rights topics. The x-axis shows the percentage of constitution sections and consultation responses. The y-axis shows topic labels.

procedures, compulsory voting, head of state powers and eligibility, judiciary structure, and municipal government. Yet it also affirms the substantial agreement across many of the top topics raised in both consultation responses and national constitutions.

Across the 184 institutional topics tracked, there is a strong positive correlation ($r = 0.75$, $p < 0.01$) between the institutional topics raised in national constitutions and Chilean consultation responses. We can thus say that the institutional topics of interest to consultation participants largely align with those that feature most prominently in national constitutions.

The above results were obtained using aggregated responses. Statistical analysis shows that within institutional topics there is a very strong positive correlation between the three administrative levels and between the three municipalities (Table 5). We can conclude that aggregation of consultation responses within institutional topics is appropriate since neither administrative level nor municipality affects response distribution for this category of topics.

## Summary

Our methodology enabled us to provide answers to our research questions as follows. First, by comparing the semantic similarity between topics and constitution sections with the semantic

**Table 4. Pearson's correlation coefficients for comparisons of consultation response topics across administrative levels and municipality types for 97 rights topics.**

| Source | Source | Pearson's |
|---|---|---|
| Regional | Provincial | 0.97 |
| Regional | Local | 0.95 |
| Provincial | Local | 0.97 |
| Provincial-Urban | Provincial-Rural | 0.97 |
| Provincial-Urban | Provincial-Mixed | 0.94 |
| Provincial-Rural | Provincial-Mixed | 0.97 |
| Local-Urban | Local-Rural | 0.99 |
| Local-Urban | Local-Mixed | 0.99 |
| Local-Rural | Local-Mixed | 0.99 |
| Provincial-Urban | Local-Urban | 0.96 |
| Provincial-Rural | Local-Rural | 0.98 |
| Provincial-Mixed | Local-Mixed | 0.93 |

Pearson's correlation coefficients are shown for a) pairwise comparisons between the distributions of consultation responses over rights topics for the three administrative levels of consultation (top three rows) and b) pairwise combinations of comparisons between the distributions of consultation responses over rights topics for municipality types. All pairs of distributions are highly correlated with $p < 0.001$.

similarity between topics and consultation responses, and then measuring the frequency of topics over constitution sections and consultation responses we found:

- The distribution of constitution sections over topics was different from the distribution of consultation responses over topics. In other words, the topics of most interest to Chilean consultation participants are not the same as the topics that are most prevalent in national constitutions globally. There are both procedural and substantive reasons why this may not be surprising. First, some of the consultation response options were predefined, thus steering at least some participant responses toward those concepts rather than to the full set of concepts that constitutions address. Second, constitutions globally address a wide range of contexts and needs that may not all be relevant in Chile which, likewise, has its own particular contexts and needs to address. What is informative about this result, though, is the discovery of exactly where and how much the priorities of Chilean participants differ from those raised in constitutions globally.

- The distribution of constitution sections over rights topics was different from the distribution of consultation responses over rights topics. In other words, the set of rights topics of most interest to Chilean consultation participants diverges from those most prevalent in constitutions globally.

- The distribution of constitution sections over institutional topics was the same as the distribution of consultation responses over institutional topics. The institutional topics of most interest to Chilean consultation participants are therefore largely aligned with the most prevalent institutional topics in constitutions globally.

Second, by measuring the semantic similarity between topics in our reference ontology and consultation responses we found:

- The distributions of consultation responses over topics are highly correlated for consultation responses collected at the local, provincial, and regional levels. The distribution of

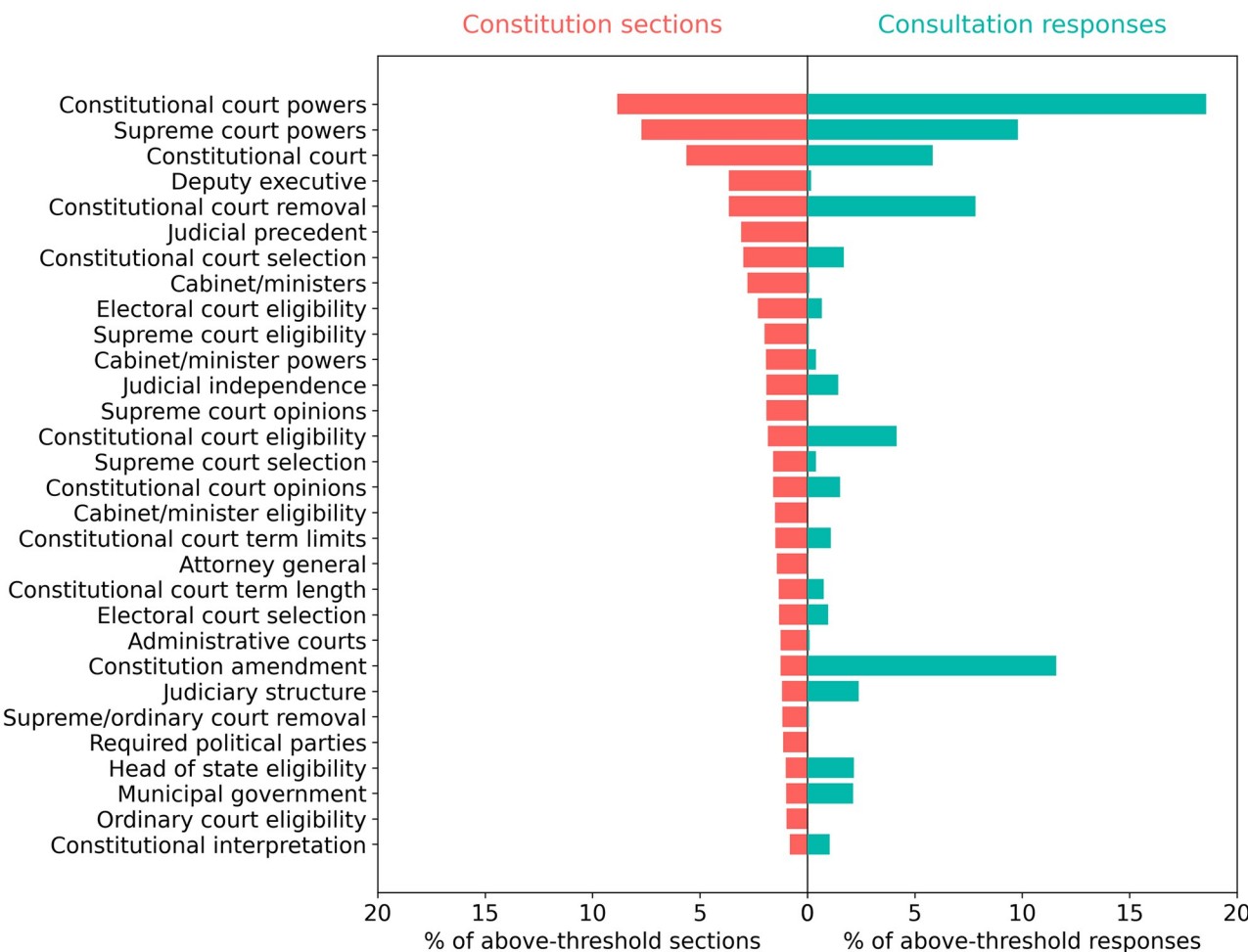

**Fig 5. Comparison of the ranked distribution of constitution sections (red) and consultation responses (green) across the 30 institutional topics that are most prevalent in national constitutions.** The figure shows the number of constitution sections and consultation responses for each institutional topic, expressed as a percentage of the total number of above-threshold sections or responses across all institutional topics. The x-axis shows the percentage of constitution sections and consultation responses. The y-axis shows the topic labels.

consultation responses over topics is thus independent of the administrative level at which consultations were conducted. This holds true for all topics, rights topics alone, and institutional topics alone.

- The distributions of consultation responses over topics are highly correlated for consultation responses collected in urban, rural, and mixed municipality types. The distribution of consultation responses over topics is thus independent of the type of municipality in which consultation participants live. This holds true for all topics, rights topics alone, and institution topics alone.

- We conclude that the slightly different structures of consultations held at the local, provincial, and regional levels and the different residential contexts of consultations held in urban, rural, and mixed municipalities did not impact the relationships between topics raised in consultation responses and those raised in constitutions.

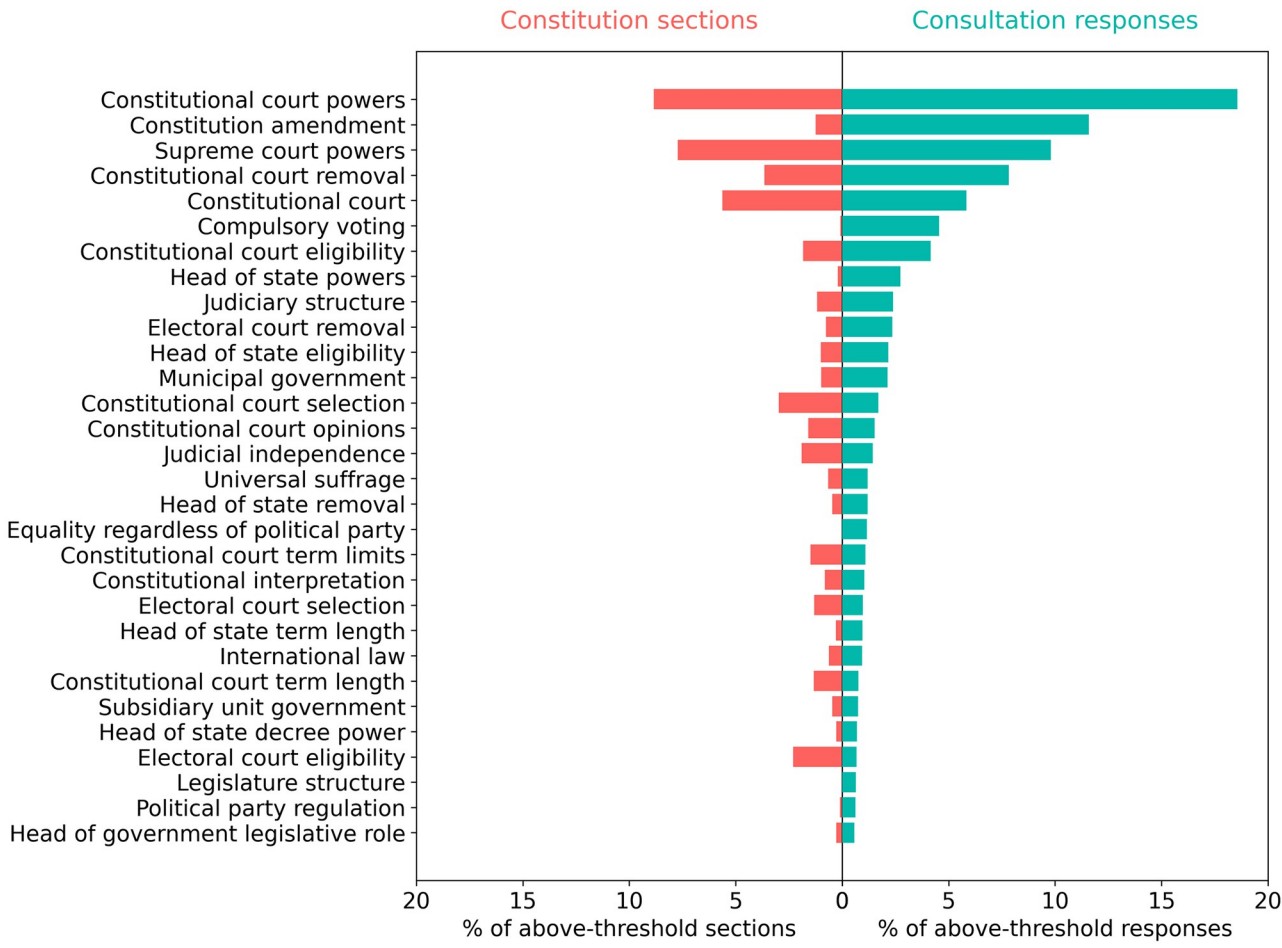

**Fig 6. Comparison of the ranked distribution of consultation responses (green) and constitution sections (red) across the 30 institutional topics that are most prevalent in consultation responses.** The figure shows the number of constitution sections and consultation responses for each institutional topic, expressed as a percentage of the total number of above-threshold sections or responses across all institutional topics. The x-axis shows the percentage of constitution sections and consultation responses. The y-axis shows topic labels.

The constitutional preferences of participating citizens as expressed in consultation responses thus differ from the topics in national constitutions, with the possible exception of institution-related topics. There is no difference between responses based on the administrative level of consultation, or the municipality type where consultations were held. In other words, participating citizens share the same constitutional preferences regardless of whether they were consulted at the local, provincial, or regional level, and regardless of whether they lived in urban, rural, or mixed areas. The evidence supports the assertion that participating citizens share a set of values, and these values expressed in this consultation differ from the topics most commonly seen in national constitutions.

## Discussion

Prior efforts to analyze the Chilean consultation data have been laudable, but these were constrained by the lack of tools and analytical frameworks suited to the scale of the textual analysis. The official analysis of the Chilean consultation data was conducted by a government-appointed "Systematization Committee" [17]. This body was headed by a group of non-

**Table 5. Pearson's correlation coefficients for comparisons of consultation response topics across administrative levels and municipality types for 184 institutional topics.**

| Source | Source | Pearson's |
|---|---|---|
| Regional | Provincial | 0.99 |
| Regional | Local | 0.94 |
| Provincial | Local | 0.94 |
| Provincial-Urban | Provincial-Rural | 0.93 |
| Provincial-Urban | Provincial-Mixed | 0.86 |
| Provincial-Rural | Provincial-Mixed | 0.90 |
| Local-Urban | Local-Rural | 0.98 |
| Local-Urban | Local-Mixed | 0.98 |
| Local-Rural | Local-Mixed | 0.99 |
| Provincial-Urban | Local-Urban | 0.94 |
| Provincial-Rural | Local-Rural | 0.89 |
| Provincial-Mixed | Local-Mixed | 0.85 |

Pearson's correlation coefficients are shown for a) pairwise comparisons between the distributions of consultation responses over institutional topics for the three administrative levels of consultation (top three rows), and b) pairwise combinations of comparisons between the distributions of consultation responses over institutional topics for municipality types. All pairs of distributions are highly correlated with $p<0.001$.

partisan academics and researchers based at the United Nations Development Programme (UNDP). Their final report identified the main topics raised by citizens in each category (values, rights, duties, and institutions), using a combination of qualitative techniques and quantitative bag-of-words methods. A team of computer scientists also conducted an enlightening early study that explored how various NLP techniques classify citizens' open-ended explanations of constitutional ideas [18]. Other analyses of public input have been similarly inductive, relying on word counting and topic modeling to generate insights in a data-driven way [19–21].

Our approach, in contrast, is a formal, confirmatory one, in which we employ a well conceptualized set of topics in comparative law to facilitate comparison. We are able to analyze more sharply the semantic meaning and conceptual content of the Chilean consultation responses by leveraging sentence-embedding techniques. Sentence embeddings represent sentences as numerical vectors in a high-dimensional semantic space—intuitively, sentences that appear in similar contexts should be closer in this space (i.e., the distance between them will be shorter). Sentence embeddings promise to convey the meaning of the entire sentence, by representing word sequences [15]. Sentence-embedding techniques are a type of transfer learning method, in which large pretrained models—such as the multilingual USE model we employ—are used for smaller context-specific tasks. These approaches are only starting to see use in political science research [22].

It is important to emphasize the advantages of our approach compared to other methods. An alternative approach such as word embeddings, for example, has recently gained traction as another way to represent political texts numerically [23,24]. These high-dimensional, word-level vectors are usually obtained via neural networks trained using millions of natural language documents such as English Wikipedia articles [25]. Word embeddings avoid the problems of bag-of-words methods that don't account for the semantic similarity of word roots and synonyms, since word embeddings place all those words closer in the language space. These word embedding methods, however, face significant problems when used to perform

classification and discovery tasks on political texts. In particular, it is unclear how to represent the meaning of a given text (speech, tweet, etc.) by aggregating its multiple word embeddings. A simple linear combination of word embeddings completely ignores the text's semantic structure—for instance, the difference between "democracy is better than dictatorship" and "dictatorship is better than democracy" gets lost. These shortcomings would be especially problematic for consultation responses, as we wish to represent and analyze the constitutional preferences of citizens as close to their original meaning as possible.

The issue of semantic representation becomes even more acute when discussing more common text-as-data methods. As noted, the Chilean Systematization Committee, the official panel in charge of analyzing the consultation data, used a standard bag-of-words approach [26] in which each text was represented by a term-frequency vector, or a series of 0s and 1s for each term in the full text corpus. One of the major quandaries of these methods is how to define what a term is. If one simply uses words, for example, "constitution" and "constitutional" would be separate terms, though they obviously refer to the same concept. Preprocessing techniques such as stemming are sometimes employed to tackle this problem of common word roots, but the results in practice are often lackluster [27]. When using sentence-level embeddings, in contrast, no such preprocessing is required—an immense advantage in using these methods to analyze processes like public consultations that produce large volumes of text. Another limitation of bag-of-words methods is related to synonyms or words with very similar meaning, such as "legislature" and "congress." Ideally, we would want to represent them in the same term, but it is unclear how to do so using the bag-of-words method. Again, sentence-level embeddings do not face similar issues, because encoding ensures that sentences conveying the same meaning using different words are close together in the semantic feature space.

By using the CCP ontology alongside sentence-level embeddings of constitution sections and consultation responses, we are able to identify the topics raised in consultation responses, their prevalence, and their similarity to (and divergence from) topics in the world's constitutions. We are also able to identify constitution sections and consultation responses that are not similar to any existing CCP topic. We call these out of vocabulary sections and responses because they fall outside the CCP vocabulary we use to classify these texts—a set of sections and responses that we do not analyze here but plan to include in our future research. This natural extension of our approach is especially promising given the limitations of purely inductive discovery via topic models. Perhaps the most pressing issue is "researcher degrees of freedom" [28]. Topic model users have to make a series of decisions—for example when preprocessing the text, selecting the number of topics, and interpreting the topics—that can dramatically alter the final results [27,29]. In the absence of theoretical guidance and validation, the worry is that topic models can be used to retrieve whatever users want them to retrieve [30,31]. We instead employ a comprehensive ontology to anchor our analysis. Any out of vocabulary excerpts can be used to enrich future versions of the ontology—analogous to a biologist discovering a species not yet in the collective taxonomy.

Our methodology uses a reference ontology and sentence-level similarity matching to deliver substantive insight. Several new themes emerge from our analysis of the 2016 constitutional consultation in Chile. One is Chilean organizers' disproportionate focus on rights in structuring fully half of the consultation questions around rights and duties, as opposed to more structural aspects of the constitution such as executive power or economic redistribution. This focus on rights makes certain sense: rights are accessible, intelligible legal devices that affect citizens directly. They are solutions that are cognitively clear and, arguably, powerful. On the other hand, institutional elements are equally consequential, and both executive power and inequality have been common critiques of Chilean politics since authoritarian rule.

The extensive focus on rights reminds us of arguments that constitutional reform in Latin America has often tinkered with changes in rights without addressing the "engine room" of the constitution—that is, executive power [32]. The limited focus on the engine room was thus as much a choice of the consultation organizers as it was participating citizens.

Accounting for the consultation structure's disproportionate focus on rights, however, another key theme emerges when we explore which rights Chilean participants and global constitutions raise most often. The most-cited rights provisions in constitutions globally range from a ban on cruel treatment to equality before the law and press freedom. Yet Chilean participants far and away prioritized equality issues—both general guarantees of equality before the law and specific guarantees of equality based on gender, sexual orientation, and skin color. This is a particularly salient finding in the context of the drafting process that would eventually unfold in Chile. The Chilean Constitutional Convention that was popularly elected in 2021 was one of the first of such institutions globally to require gender parity and among the few ever to prioritize indigenous representation, reserving 17 seats (roughly 11% of all seats) for indigenous peoples.

Another key theme to highlight is the shared focus of Chileans on a similar set of issues, regardless of the urban or rural contexts where respondents live. Further, this shared set of issues was expressed regardless of whether consultations were convened by community groups at the local level or by government officials at the provincial and regional levels. This uniquely Chilean set of constitutional priorities becomes even more clear when compared to constitutional ideas already entrenched in national constitutions worldwide. This comparison makes clear that the Chilean constitutional priorities expressed in these consultations emphasize different topics from the ideas in current national constitutions—a finding of critical import for drafters in Chile, as everywhere, to consider. It may be, of course, that consultation participants focused on novel ideas because they took established constitutional ideas for granted. Either way, the unique focus of consultation participants provides valuable information for constitution drafters. Our methods provide new tools to draw such themes from the unwieldy and unstructured but critically important data produced by what continues to be one of the most ambitious constitutional projects in modern times.

Our approach is based on what we call conceptual ecology. Analogous to a biologist's estimate of the prevalence of a set of species across different habitats, conceptual ecology involves the identification and census of concepts in various populations. Thus, the set of comments generated by public consultation represents a particular population within which to identify and catalog concepts. Systematizing this analysis across populations generates comparable reference points that are of descriptive and analytic interest to scholars, constitution drafters, and other groups invested in the constitutional preferences of the public.

Our fundamental ecological question—"Which constitutional topics are of most interest to the public?"—is intractable without an applicable and comparable ontology. Ontologies are crucial intellectual tools for organizing and representing domains of knowledge, enabling the classification, tracing, and comparison of entities in a given domain, whether animals, medical conditions, or social science concepts. We employ a constitutional ontology designed to index the conceptual content of the world's constitutions.

The benefit of connecting consultation input to constitutional texts is significant. At a functional level, it gives constitution drafters a range of examples of how current constitutions address a topic raised by their constituents. If we learn, for example, that citizens are especially interested in constitutionalizing protection for the environment, for example, we may want to compile and review relevant passages in the world's constitutions so drafters can consider their options concerning a major priority expressed by citizens. But at a more conceptual level, it also enables us to identify the differences between the public's constitutional preferences in a

given country and the constitutional ideas currently embodied in the world's higher laws. Drafters could use this information to emphasize the unique priorities of their citizens.

## Conclusion

For years, analysts of public consultations—constitutional and otherwise—have struggled to sort through voluminous public input to deliver actionable intelligence to decision-makers.

Intelligence that allows leaders to make (and dispute) arguments on behalf of citizens. But how to analyze and represent the hundreds of thousands of citizen preferences expressed in these consultations in a way that is useful to drafters?

Our approach is predicated on the existence of reference ontologies that represent some part of the conceptual scope of a domain. We begin with a comprehensive ontology developed for the interpretation of historical constitutions that is widely used by comparative constitutional scholars. To match citizen suggestions with topics in this ontology, we build on advances in natural language processing, specifically sentence-level embeddings. We demonstrate that by using a reference ontology we can assess the conceptual alignment of textual corpora, in this case, constitutional texts and public consultation input from Chile. The insight into the constitutional preferences of citizens that conceptual alignment provides has the potential to increase the reach and impact of public input that too often goes under-utilized. At the same time, drafters and advocates are provided with actionable data on the constitutional preferences of constituents.

The methods we develop here produce the evidence that would enable delegates to make informed claims about the relative interests of the public. The distribution of these interests can also be compared at discrete administrative levels, or any other crosscuts of the public for which data are available. In Chile, for example, equality regardless of sexual orientation and gender appear high on the list of topics mentioned by citizens, and the overall distribution of rights topics does not exhibit meaningful variation across administrative levels. This kind of information can be helpful in setting constitutional priorities. Importantly, it also allows drafters to evaluate competing claims about citizen preferences, a common rhetorical tactic [33]. All of this gives the opinions of citizens a seat at the constitution-drafting table.

What's more, our methods can empower non-state actors dedicated to improving transparency and holding delegates accountable during the process of constitution-making. Drafters can use the evidence produced by our methods to assess each other's claims about constitutional preferences. Similarly, scholars, journalists, and activists have the opportunity to evaluate independently whether the interests of the citizenry are being represented properly, or perhaps manipulated. As Lindhal [34] reminds us, "there can be no gathering together of a multitude into a collective subject without acts that seize the initiative to include and exclude." Insofar as drafters, or any state actor, seek exclusion, civil society can use evidence of citizen desires and aspirations, produced by our methods, as a counterweight toward inclusion and pluralism.

The analysis, therefore, provides scholars, citizens, and decision-makers with an inventory of the ideas that Chileans would like to see in their constitution. We know that this citizen focus on rights may not necessarily be the result of organic preferences so much as the organizers' method of consultation and line of questioning. The implication is that it may be just as important to focus on the design of the consultation exercises as on the analysis of the results.

In this paper we have described the application of our methodology to Chilean consultation responses. However, the same methodology can be applied to any textual corpora in several languages using any number of reference ontologies in the same or different languages (for example, specialized ontologies related to gender issues or climate justice). For example, we

are currently using the process to extend our analysis by tracing and comparing topics raised across public consultations, drafting body deliberations, and draft constitutions in Chile's sequential efforts to reform its constitution in 2015–17, in 2019–22, and in 2023. This enables us to assess the topics raised in Chilean consultations and drafting body deliberations, whether drafters leveraged public input rhetorically or substantively in the drafting process, and how the design and timing of public consultations shaped the draft constitutions. In addition, the methodology can be used to directly compare ontologies within and across languages. As such, the methodology has far-reaching implications for coordinating concepts and data across researchers and practitioners, and for supporting the generation of research questions and testable hypotheses in a number of domains.

## Supporting information

**S1 Appendix. Topics in the CCP ontology.** Below is a representation of the 334 topics in the CCP ontology, organized into 12 categories.
(TIF)

**S2 Appendix. Replication data in repository.** The repository contains the following folders and files to support replication of our results in this paper: data (CCP ontology, CCP institution topics, CCP rights topics, municipality types, and predefined consultation topics), outputs (distribution data and municipality data that are calculated spreadsheets containing counts, scaled values, and percentages generated from the main similarity scores and used in data analysis), proc (main similarity scores), Jupyter notebook used to build calculated spreadsheets from the main similarity scores, and Jupyter notebook used for data analysis.
(TIF)

## Author Contributions

**Conceptualization:** Zachary Elkins, Matthew Martin, Ashley Moran.

**Data curation:** Zachary Elkins, Roy Gardner, Ashley Moran.

**Formal analysis:** Roy Gardner.

**Funding acquisition:** Ashley Moran.

**Methodology:** Andrés Cruz, Roy Gardner, Matthew Martin, Ashley Moran.

**Project administration:** Matthew Martin.

**Software:** Andrés Cruz, Roy Gardner.

**Supervision:** Matthew Martin.

**Validation:** Andrés Cruz, Roy Gardner, Matthew Martin.

**Visualization:** Andrés Cruz, Roy Gardner, Ashley Moran.

**Writing – original draft:** Andrés Cruz, Zachary Elkins, Roy Gardner, Matthew Martin, Ashley Moran.

**Writing – review & editing:** Andrés Cruz, Zachary Elkins, Roy Gardner, Matthew Martin, Ashley Moran.

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
