## [Decision Letter · Decision Letter 0]

29 Aug 2023

PONE-D-23-21004What kind of constitution do the people want? A new method for analyzing public consultation dataPLOS ONE

Dear Dr. Martin,

Thank you for submitting your manuscript to PLOS ONE. After careful consideration, we feel that it has merit but does not fully meet PLOS ONE’s publication criteria as it currently stands. Therefore, we invite you to submit a revised version of the manuscript that addresses the points raised during the review process.

One reviewer is rather critical of your manuscript and considers the limitations / gaps to be not easily fixable. Two reviewers have a very positive view of the paper, but they also name points that need more and more convincing discussion. I am siding with the latter two reviewers in that I see a path forward for your paper, but I urge you to take all reviewers' comments seriously in the revision. One reviewer request concerns providing more context about the constitution-making process in Chile. In that respect, the paper by Larrain et al. in Public Choice (2023) could be helpful. I would ask you, and this relates to concerns already voiced by the reviewers, to clarify what exactly you are measuring here and how policy implications depend on what your data reflects. Leaving aside for the moment that the question process was apparently guided, do you measure knowledge off constitutions and what topics they typically deal with? Do you measure what topics are particularly salient in the public debate? Are you measuring actual priorities in the design process as perceived by citizens? To what extent can we really infer from your measures "what the people want", as you claim in your title? Moreover, please make sure that notes on all figures and tables clarify what exactly they represent without consulting the text. Finally, please report information about statistical significance tests consistently and correctly. Writing things like "(significant at < 0.001)" is meaningless. First of all, you fail to refer to the p-value, second you could simply name the p-value instead of saying it is smaller than 0.001 (except where you are referring to multiple p-values). Of course you can say it is significant because it is smaller than 0.001, but then you would have to apply the same threshold throughout the manuscript or provide a reason why significance should be evaluated at different levels for different tests. Later you report "significant at <0.01" and "significant with a p-value < 0.004" - that is just not how significance tests work and are reported. The easiest fix would be to simply use the 1%-threshold for all significance tests in the paper and report when this threshold is met (or to just name the exact p-values rather than below what threshold they are).

We look forward to receiving your revised manuscript.

Kind regards,

Jerg Gutmann

Academic Editor

PLOS ONE

5. We notice that your supplementary figures are uploaded with the file type 'Figure'. Please amend the file type to 'Supporting Information'. Please ensure that each Supporting Information file has a legend listed in the manuscript after the references list.

Reviewers' comments:

Reviewer's Responses to Questions

**Comments to the Author**

1. Is the manuscript technically sound, and do the data support the conclusions?

Reviewer #1: Yes

Reviewer #2: No

Reviewer #3: Yes

2. Has the statistical analysis been performed appropriately and rigorously? 

Reviewer #1: Yes

Reviewer #2: No

Reviewer #3: Yes

3. Have the authors made all data underlying the findings in their manuscript fully available?

Reviewer #1: Yes

Reviewer #2: Yes

Reviewer #3: Yes

4. Is the manuscript presented in an intelligible fashion and written in standard English?

Reviewer #1: Yes

Reviewer #2: No

Reviewer #3: Yes

5. Review Comments to the Author

Reviewer #1: Chile makes for an interesting data point on alignment of popular and professional const drafting preferences.

- is there a strategy / way forward for generalizing this / incorporating more cases so that claims can be made about this alignment generally as opposed to just the Chilean case?

- very descriptive, which is fine given how early we are in use of such tools in con law, but what about use of covariates to facilitate testing of hypotheses about why there is / isn’t alignment? This has been attempted before and should not be difficult to implement in topic modeling packages. Or at least articulation off hypotheses for future research?

- conclusion 1, Chilean interest isn’t the same as global interest: can you really say this given (as acknowledged earlier) that the public was guided by prompts from the drafters?

P. 29, same problem: can you really say Chileans disproportionately focused on rights, given the drafters’ prompts? You can say they focused disproportionately on equality rights, OK, but rights generally?

Reviewer #2: 1. The research design of the study can be improved substantially:

- To enhance the quality of this study, authors should refine the research questions. This refinement will greatly assist the authors in establishing a distinct theoretical framework and a robust empirical strategy. I strongly recommend concentrating on either the questions outlined within the "consultation structure" or those in "residential context" sections. It is advisable to refrain from making any causal claims due to the limitations of the available data, as asserting any form of causality based on the existing data is not feasible.

- The current work lacks a citation of relevant literature on the text-as-data approach within legal scholarship. To address this, the authors could consider starting their exploration with the comprehensive study conducted by Frankenreiter and Livermore (2020), which offers an insightful survey of computational methodologies in the realm of law. This survey notably encompasses research close to the present study in terms of substantive content. Furthermore, it is advisable for the authors to actively seek out recent studies that bear resemblance to their research in terms of methodology, such as the work by Nyarko (2019).

- The paper is deficient in terms of a coherent theoretical framework. While various concepts are introduced, they are presented without robust substantiation. This situation could be a result of absence of a well-defined research question. It is strongly advisable for the authors to rework and clarify their research question, subsequently constructing a theoretical framework that can serve as a foundation for generating empirically testable hypotheses.

2. The precise assessment of the empirical strategy employed in this study is impossible due to the absence of well-defined hypotheses for investigation.

- The rationale behind the selection of global constitutions as a basis for comparison remains unexplained. A potentially more justifiable approach could involve examining consultations across distinct geographical regions and varying levels of engagement. As it stands, the current empirical evidence is notably disorderly and fails to yield any novel insights.

- The assertion regarding the paper's methodological contribution is inaccurate. The methodology used in the paper was originally introduced by Yang et al. in 2019. Considering the swift progression in the field of large language models, it becomes apparent that the model employed in this paper is not the most current iteration available. Even granting the assumption that the model used in this paper is of high quality, a pressing question arises: why is this specific model suited for the given task? The authors neglect to address this inquiry satisfactorily. They omit a comparative analysis with other existing models and fail to present a theoretical rationale for the efficacy of their chosen model for the task at hand. Furthermore, the authors provide no validation for the reliability of their dataset.

3. Data used in this study is publically available.

4. The language used in the paper can be improved substantially.

Many sentences are ambiguous and unclear. Examples:

- The country’s 2016 consultation process was similar to that of other contemporary constitutional projects and may be something of a model case, given the regional influence of Chile and the historic nature of its reform

Instead of relevant terminology, new concepts were developed. Examples:

- Constitutional Ontology instead of ground truth.

Reviewer #3: I think this is an important contribution to the comparative constitutional studies literature. I am not a data scientist so hard to judge the methods there. The only suggestion I have is that the paper should maybe give a bit more context on the Chilean process. E.g., gender equality is one of the main topics in public consultation, but it was also the first constituent assembly with gender parity. More generally, some reflection on why equality is so high on the agenda would be useful. Maybe use the discussion for some of that? Overall, a very useful contribution to the literature.

6. PLOS authors have the option to publish the peer review history of their article (what does this mean?). If published, this will include your full peer review and any attached files.

Reviewer #1: No

Reviewer #2: No

Reviewer #3: **Yes: **Mila Versteeg

---

## [Author Response · Author response to Decision Letter 0]

5 Oct 2023

We thoroughly appreciate the detailed and helpful comments received from the editor and reviewers. We note in our "Response to reviewers" document how we sought to address each comment and where those changes can be found in the manuscript.

---

## [Decision Letter · Decision Letter 1]

20 Nov 2023

Measuring constitutional preferences: A new method for analyzing public consultation data

PONE-D-23-21004R1

Dear Dr. Martin,

We’re pleased to inform you that your manuscript has been judged scientifically suitable for publication and will be formally accepted for publication once it meets all outstanding technical requirements. Reviewer 3 suggested acceptance of your manuscript as submitted. Reviewer 1 suggested acceptance of your manuscript after revision. Unfortunately, Reviewer 2 was not able to review the revised version of the paper in time. Given the positive evaluations by Reviewers 1 and 3 and my own reading of your significantly improved version of the manuscript, I am positive that it will make an excellent contribution to PLOS ONE. However, note that there are still some critical remarks by Reviewer 1 that you might want to take a careful look at. Thank you for considering us as an outlet for your research.

Kind regards,

Jerg Gutmann

Academic Editor

PLOS ONE

Additional Editor Comments (optional):

Reviewers' comments:

Reviewer's Responses to Questions

**Comments to the Author**

1. If the authors have adequately addressed your comments raised in a previous round of review and you feel that this manuscript is now acceptable for publication, you may indicate that here to bypass the “Comments to the Author” section, enter your conflict of interest statement in the “Confidential to Editor” section, and submit your "Accept" recommendation.

Reviewer #1: (No Response)

2. Is the manuscript technically sound, and do the data support the conclusions?

Reviewer #1: Yes

3. Has the statistical analysis been performed appropriately and rigorously? 

Reviewer #1: Yes

4. Have the authors made all data underlying the findings in their manuscript fully available?

Reviewer #1: Yes

5. Is the manuscript presented in an intelligible fashion and written in standard English?

Reviewer #1: Yes

6. Review Comments to the Author

Reviewer #1: The authors have made many meaningful improvements to the paper in terms of both presentation and content. I have to say though that I'm not really sold on the answer to "how do we know that Chilean attention to certain types of provisions isn't a function of how the consultations were structured/what questions they were asked, rather than what they wanted"?

To their credit, they don't hide that this is a possibility. In fact, some of the new info they provide suggests it may be an even bigger problem/possibility: we learn that the participants were fed 44 concepts in the area of rights, more than double what they were given in the area of institutions (p. 10). Jury instructions are assumed to influence lay decisionmakers: you tackle the agenda you are given and the questions you are posed. So why isn't this different? So it still begs the question: what does this paper tell us about what exactly it is that is diverging from the content of national constitutions generally?

Some cross-country comparison of popular input into constitution-making preferences might support the hypothesis that popular preferences diverge from national constitutional content. (Maybe Iceland?) The revisions suggest though that the authors want to dig deeper into the Chilean case instead rather than going cross-country. That's fine, and understandable given the difficulty of finding a similar const-making scenario that allows so much observation of popular preference, but it may still leave readers wondering what the takeaway is for a comparative con law audience. W/r/t distribution of topics w/in the category of rights, the authors suggest:

"This could be an indication that these topics are uniquely critical to the current Chilean context."

Yes this is plausible, but it would help if they could say more about what some other explanations might be, and what's more plausible and why. For example, if you're talking about trial rights, doesn't that imply/assume some degree of familiarity with criminal/trial procedure? Same might be said of lack of attention to institutional issues as opposed to rights issues. Are we seeing divergence partly for reasons of expertise? What else do we need to consider?

If they want to stick with n=1, one strategy might be to dive into the unstructured ("self-defined") responses that make up 14% of the total suggestions. Even then, one might say, well, you've given them so many prompts in the rights area that of course they are going to dwell on rights. If I give a child a box of building blocks and there are 2x as many red blocks as any other color, then you can expect to get a lot of red structures. Another problem with the self-defined responses is that they might plausibly be more focused on what the organizers failed to include than on what the participants considered most important (gap-filling rather than preference-revealing). In other words, the structure may still be structuring the unstructured responses.

7. PLOS authors have the option to publish the peer review history of their article (what does this mean?). If published, this will include your full peer review and any attached files.

Reviewer #1: No

---

## [Editor Report · Acceptance letter]

4 Dec 2023

PONE-D-23-21004R1 

Measuring constitutional preferences: A new method for analyzing public consultation data 

Dear Dr. Martin:

I'm pleased to inform you that your manuscript has been deemed suitable for publication in PLOS ONE. Congratulations! Your manuscript is now with our production department. 

Kind regards, 

on behalf of

Prof. Dr. Jerg Gutmann 

Academic Editor

PLOS ONE